

# k-Clique counting on large scale-graphs: a survey

Büşra Çalmaz and Belgin Ergenç Bostanoğlu

Computer Engineering, Izmir Institute of Technology, Izmir, Turkey

## ABSTRACT

Clique counting is a crucial task in graph mining, as the count of cliques provides different insights across various domains, social and biological network analysis, community detection, recommendation systems, and fraud detection. Counting cliques is algorithmically challenging due to combinatorial explosion, especially for large datasets and larger clique sizes. There are comprehensive surveys and reviews on algorithms for counting subgraphs and triangles (three-clique), but there is a notable lack of reviews addressing k-clique counting algorithms for k > 3. This paper addresses this gap by reviewing clique counting algorithms designed to overcome this challenge. Also, a systematic analysis and comparison of exact and approximation techniques are provided by highlighting their advantages, disadvantages, and suitability for different contexts. It also presents a taxonomy of clique counting methodologies, covering approximate and exact methods and parallelization strategies. The paper aims to enhance understanding of this specific domain and guide future research of k-clique counting in large-scale graphs.

## INTRODUCTION

The enumeration of cliques has a complex computational nature in graph mining. The clique is a subgraph with an edge between every pair of vertices. With the aim of clique counts, cohesive substructures can be identified within graphs. The combinatorial explosion is the main challenge of clique counting algorithms due to exponential growth in possible cliques as the graph size increases.

Clique counting has many application areas: It helps to reveal cohesive structures within complex networks. For example, clique counting algorithms are used in social network analysis to identify tightly connected groups of people (*Faust, 2010*; *Han, Pei & Kamber, 2006*; *Holland & Leinhardt, 1977*; *Pan et al., 2023*; *Schank, 2007*; *Foucault Welles, Van Devender & Contractor, 2010*; *Son et al., 2012*; *Tsourakakis et al., 2011*). This facilitates community detection (*Lu, Wahlström & Nehorai, 2018*) and the understanding of social dynamics. In biological network analysis, cliques are used to identify functional modules within protein-protein interaction networks (*Betzler et al., 2011*; *Pržulj, Corneil & Jurisica, 2004*; *Saha et al., 2010*) and the identification of correlated genes (*Presson et al., 2008*). This sheds light on biological pathways and disease mechanisms such as

Corresponding author
Belgin Ergenç Bostanoğlu,
belginergenc@iyte.edu.tr

epilepsy prediction (*Iasemidis et al., 2003*). Recommendation systems also use clique analysis to identify cluster users with similar preferences (*Vilakone et al., 2018*) to enhance personalized recommendations. In fraud detection applications, the collusive groups of actors who engage in fraudulent activities (*Yu et al., 2023*) can be detected with the help of clique counts. Moreover, clique counting finds applications in diverse fields, such as graph compression (*Buehrer & Chellapilla, 2008*) and clustering (*Duan et al., 2012*). Applying it to a wide range of real-world problems illustrates the significance of clique analysis.

Comprehensive surveys in the literature (*Ribeiro et al., 2021*; *Al Hasan & Dave, 2018*; *Ortmann & Brandes, 2014*) provide in-depth insights into subgraph counting and triangle (three-clique) counting algorithms in the literature. However, there needs to be more literature that provides a comprehensive review of algorithms for counting k-cliques for k is greater than three. The existence of larger cliques suggests the presence of a more robust and cohesive groups, which can be beneficial in identifying tight-knit communities or clusters. In particular, finding cohesive groups in domains such as social networks or biological networks facilitate the comprehension of the underlying organization of the network, as they can reveal more intricate and complex structures in the data. It has been demonstrated that larger cliques are more stable and reliable clusters than smaller ones. This is due to their greater resilience to noise and outliers, which allows for more accurate and meaningful groupings.

This survey's primary objective is to provide a theoretical understanding of cliques within graphs and facilitate the development of more robust and scalable algorithms. This study seeks to illuminate effective strategies for addressing this fundamental problem in graph analysis by examining existing approaches and identifying their strengths and weaknesses. The goal of this work is to enhance both theoretical research and practical applications in the field of graph analysis. Our results will contribute to developing more efficient algorithms capable of dealing with more intricate network structures.

First, the survey methodology is explained by detailing the study's purpose, research questions, data sources, and literature reviewing strategies. Then, it defines terms related to basic principles and clique counting techniques. Two essential algorithms (*Bron & Kerbosch, 1973*; *Chiba & Nishizeki, 1985*) form the basis of most clique counting algorithms. This paper gives an overview of these algorithms as the baseline for better understanding before analyzing other algorithms. After this introduction, a systematic evaluation provides insights into the papers, including their practical applications and performance across different graphs. We emphasize performance metrics, scalability, and practical applications to understand the effectiveness of different clique counting methods. Finally, based on the results presented in their respective papers, we discuss the algorithms. The aim is to provide a guide for future scalable and efficient clique counting algorithms.

## PRELIMINARIES

This section presents concepts and terminology associated with clique counting that will be employed throughout this article.

**Graph**: A graph, denoted as $G(V, E)$, comprises a collection of vertices (or nodes) represented by $V$ and a set of edges denoted by $E$, which establish connections between

vertices. A **directed graph** consists of vertices connected by edges, where the edges have a specific direction, indicating a one-way flow of information or relationship between the connected nodes. There is no such direction in the **undirected graph**. Let $n = |V|$ and $m = |E|$ represent the number of nodes and edges, respectively, in the graph G. The degree (denoted as $d(u)$) of a node $u$ is the number of its neighboring vertices, representing the count of adjacent edges connected to that node. A k-graph refers to a graph with $k$ nodes. In a graph, a **path** is a series of interconnected vertices where each one is linked to the next by an edge. A graph is **connected** if any two vertices can be linked via a path and unconnected if there are pairs of vertices with no path between them. A **simple, undirected graph** has no self-loops or multiple edges between the same two vertices. In graph theory, small, connected subgraphs are called graphlets or motifs. A graph is known as a **tree** when it is connected and has no cycles, which means there is only one path between any two vertices. On the other hand, a collection of trees that are not necessarily interconnected is known as a **forest**. The minimum number of forests required to partition the graph's edges is the **graph's arboricity**.

**Subgraph**: A graph $G(V, E)$ has a subgraph $G_s(V_s, E_s)$, if $V_s$ is a subset of $V$ and $E_s$ is a subset of $E$. An **induced** subgraph is a subset of the original graph created by selecting a set of vertices and including all connecting edges. It resembles a snapshot of the original graph, emphasizing only the chosen vertices and their interconnected edges.

**Subgraph isomorphism**: Consider two graphs, $G$ and $H$. An isomorphism between the two exists when there is a one-to-one correspondence, or bijection, called f, between their vertices ($f : V(G) \rightarrow V(H)$). For an isomorphism to be valid, any two vertices, u and v, must be adjacent in $G$ if their corresponding vertices, $f(u)$ and $f(v)$, are adjacent in $H$. When graphs meet this criterion, they are considered isomorphic and are said to be topologically identical.

**Degeneracy**: The degeneracy $d$ is the smallest integer, so every subgraph within the graph contains at least one vertex with a degree not exceeding $d$.

**Directed acyclic graph (DAG)**: A DAG is a graph in which loops do not exist, i.e., it is impossible to follow a sequence of edges and return to the same vertex.

**Clique**: A clique is a maximal subgraph where every pair of vertices is connected by an edge, defining a complete subgraph. A k-clique refers to a clique with $k$ nodes.

**Maximal clique**: A maximal clique is a clique that cannot be expanded by including one more adjacent vertex, meaning it is not a subset of any larger clique in the graph.

**Bi-clique**: A bipartite graph is a graph where the vertices can be divided into two distinct, non-overlapping subsets if every vertex of the first set is connected to every vertex of the second set. Such a graph is called a **complete bipartite graph** or **bi-clique**.

**Quasi-clique**: A quasi-clique is a subset of vertices in a graph where each vertex is connected to a substantial proportion of the other vertices in the subset, although not necessarily all. In contrast to a complete clique, where every vertex is connected to every other vertex, a quasi-clique allows some vertices to have fewer connections within the subset. Figure 1 illustrates various types of cliques.

**Graph orientation**: The fundamental step of graphlet mining algorithms involves a preprocessing stage to orient the graph. The objective is to establish an acyclic orientation

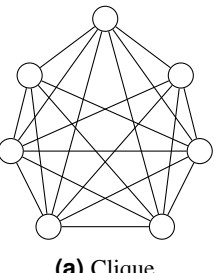 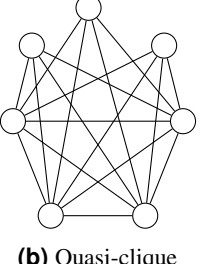 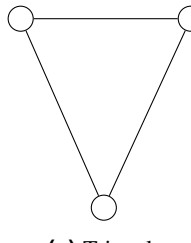 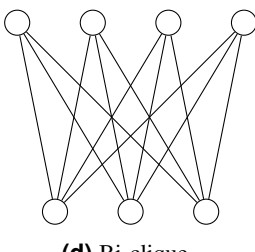

**(a)** Clique  **(b)** Quasi-clique  **(c)** Triangle  **(d)** Bi-clique

**Figure 1** **(A–D) Clique types.**

of an undirected graph, whereby each edge is assigned a direction that avoids the formation of directed cycles and any potential duplication of subgraph determinations. Three primary approaches are employed to achieve this: degree orientation, degeneracy orientation, and color orientation. In **degree orientation** techniques, nodes are arranged in order of priority based on their degrees. The higher probability is assigned to higher-degree vertices. Then, the edges are oriented from vertices with low priority to high priority. On the other hand, **degeneracy orientation** uses an iterative process of removing vertices with the minimum degree. The minimum-degree vertex is eliminated, and all remaining vertices' degrees are updated. The priority is assigned based on removal time. In **color orientation** technique, a greedy coloring algorithm (*Hasenplaugh et al., 2014*; *Yuan et al., 2017*) is applied to color the graphs with $m$ colors. Each node is assigned a color between $1, \ldots, m$ in this method, but it is ensured no two adjacent nodes have the same color. Based on this ordering, the graph is oriented to construct a directed acyclic graph (DAG).

**Edge density**: The edge density of a graph quantifies the ratio of existing edges to the total number of possible edges within the graph. It is computed by dividing the number of edges in the graph by the total number of possible edges. This ratio is expressed mathematically as $\frac{m}{\binom{n}{2}}$, where $m$ represents the number of edges present in the graph and $n$ is the number of vertices.

## SURVEY METHODOLOGY

This section outlines the purpose and audience of the study and presents a taxonomy of the methods based on the research questions. A literature review methodology is detailed to ensure the comprehensiveness of this survey, along with the data sources from which the articles were collected. The criteria for the inclusion or exclusion of articles for scoping this research are also explained in the following section.

This paper delves into algorithms for counting k-cliques within a given graph $G$. Clique counting is challenging due to the exponential growth in possible combinations, particularly as the clique size $k$ increases. The study surveys a range of methodologies, spanning from exhaustive enumeration to approximation techniques, to provide a comprehensive overview of the strategies used in clique counting. Our research explores algorithms that can efficiently count simple connected, non-isomorphic subgraphs on a single graph.

### The purpose, importance, and audience of this survey

This survey aims to provide a comprehensive overview of the k-clique counting algorithms. Therefore, extensive literature reviews have been conducted to thoroughly understand the evaluation of k-clique counting algorithms, aiming to identify their gaps, strengths, and limitations for facilitating future research paths.

Despite the availability of comprehensive surveys that provide in-depth insights into subgraph counting and triangle (three-clique) counting algorithms (*Ribeiro et al., 2021*; *Al Hasan & Dave, 2018*; *Ortmann & Brandes, 2014*), there is a notable gap in the literature concerning a thorough review of algorithms for counting k-cliques where *k* is greater than three. This survey aims to fill that gap, addressing the need for a comprehensive review of k-clique counting algorithms beyond triangles. The challenges of this problem generally limit the size of cliques used in various applications to smaller ones. However, larger cliques can offer different insights and can serve as more robust choices for clustering, classification, or detection algorithms (*Duan et al., 2012*; *Vishwanathan et al., 2010*; *Lu, Wahlström & Nehorai, 2018*; *Yu et al., 2023*). By providing a detailed review of k-clique counting algorithms, this survey aims to support the development of more efficient and practical techniques for analyzing large and complex datasets.

The researchers and practitioners in network analysis, data mining, bioinformatics, and any field where understanding the structure of complex networks is crucial can be the audience of this study.

### Research questions

In this survey, the following research questions are formulated considering the research objectives:

- What are the challenges of exact clique counting algorithms on large and complex datasets?
- Which strategies are employed to facilitate the exact clique counting on large-scale datasets?
- Why is there a need for approximate clique counting algorithms?
- How to approximate clique counting techniques scale with increasing graph size and complexity?
- What is the common preprocessing step in clique counting algorithms to eliminate duplicate exploration?
- Which parallelization strategies are used to effectively improve the performance of clique counting algorithms on large-scale graphs? How do they integrate parallelization strategies with the clique counting algorithms?

The paper critically evaluates clique counting algorithms to reveal their strengths, weaknesses, and applicability across diverse scenarios, enhancing our understanding of this fundamental problem in graph analysis. A carefully constructed taxonomy (Fig. 2) that systematically categorizes the approaches used in clique analysis is presented in light of these

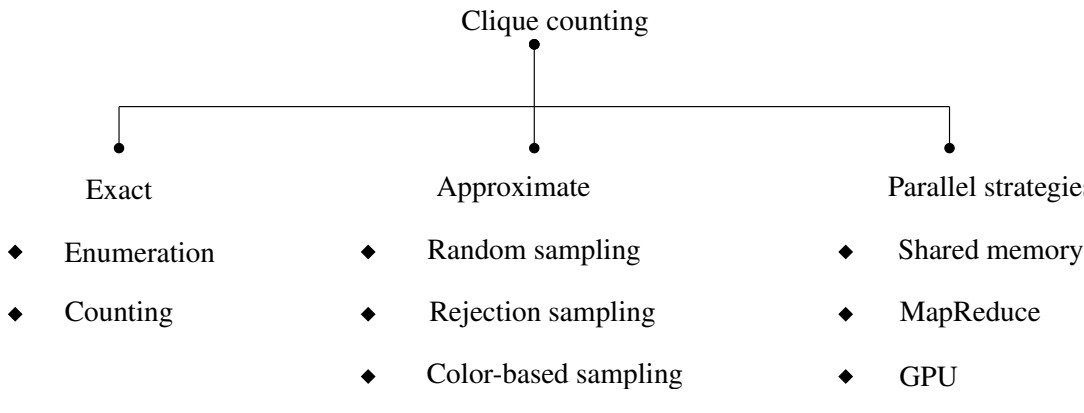

**Figure 2** A taxonomy of k-clique counting algorithms.

research questions and the evolution of clique counting algorithms. The methodologies for clique identification and enumeration are categorized based on their level of precision.

Exact techniques meticulously identify all cliques or count their existence within a graph, whereas approximate counting methods provide estimations that are especially useful for analyzing large, intricate graphs. The research objectives and the graph's scale determine the methodology choice. Exact techniques are well-suited for smaller or medium-sized graphs, while approximate counting is advantageous for exploring larger network structures. Within the scope of this study, the investigated approximate approaches rely on different sampling strategies, including random sampling, rejection sampling, and color-based sampling. Parallelization strategies delineate how computation is distributed locally across processors or globally across distributed systems. This structured taxonomy provides a framework for understanding and organizing clique counting techniques, fostering a deeper comprehension of the methodologies employed in clique analysis research.

## Data sources and literature search methodology

The articles analyzed in this research were collected from Google Scholar, IEEE Xplore, WoS, Arxiv, ACM Digital Library, SIAM, SpringerLink, and other search engines using the keywords "clique counting," "clique enumeration," and "k-clique counting." The articles were selected manually from these sources by scanning the title, keywords, and abstract. In addition, a snowballing technique was used to include relevant publications cited by the selected articles. Table 1 presents the search engines used, their respective links, and the search keywords.

## Criteria for inclusion or exclusion

Many algorithms count or estimate graphlets involve counting cliques of sizes 3, 4, and 5. However, these algorithms are beyond the scope of this paper because their primary focus is not on cliques directly. Instead, they aim to count or predict k-graphlets, all graphlets formed by k nodes, efficiently, typically with a limited value of k such that $k \leq 5$ due to combinatorial explosion (*Ahmed et al., 2015*; *Wang et al., 2017*; *Pinar, Seshadhri & Vishal, 2017*; *Rossi, Zhou & Ahmed, 2018*; *Bressan, Leucci & Panconesi, 2019*).

**Table 1** The search engines used, their respective links, and the search keywords.

| | Links | keywords |
|---|---|---|
| Google Scholar "k-clique counting" | https://scholar.google.com/ | "clique counting," "clique enumeration," |
| ACM Digital Library | https://dl.acm.org/ | * |
| IEEE Xplore | https://ieeexplore.ieee.org/ | * |
| WoS | https://www.webofscience.com/ | * |
| Arxiv | https://arxiv.org/ | * |
| SIAM Journal | https://www.siam.org/ | * |
| SpringerLink | https://link.springer.com/ | * |

**Notes.**
*The keywords indicated in the first row are used for all search engines.

Moreover, it is essential to note that this study does not analyze triangles(three-cliques), bi-cliques, and quasi-cliques. These clique types offer unique insights into graphs' underlying structure and connectivity (*Pagh & Tsourakakis, 2012*; *Sanei-Mehri, Sariyuce & Tirthapura, 2018*; *Jain & Seshadhri, 2020b*). Triangles represent the simplest clique form with three vertices. On the other hand, bi-cliques have a bipartite structure, comprising two distinct sets of vertices with complete interconnections between them, unlike k-cliques involving a single set of vertices. Quasi-cliques relax the connectivity requirement, allowing subsets of vertices to exhibit significant pairwise connections without necessitating complete subgraphs. While k-cliques focus on complete subgraphs of a specific size, these alternative clique types, such as triangles, bi-cliques, and quasi-cliques, reveal diverse connectivity patterns within graphs. The clique counting methods on dynamic graphs that change over time are not within the scope of this paper.

This study focuses exclusively on algorithms that count k-cliques. In the examined papers, numerous references were found regarding algorithms for enumerating/counting maximal cliques, which can be adapted for clique counting. If a maximal clique enumerating/counting algorithm has contributed to developing a k-clique counting algorithm, served as its foundation, or evolved into a version of a k-clique counting algorithm, it has been included in this study. Consequently, these articles are included within the scope of this paper. In this context, only peer-reviewed conference papers and journal articles in English that address counting k-cliques are included. Non-peer-reviewed articles, such as preprints, theses, dissertations(except one (*Jain, 2020*)), and articles written in languages other than English, are excluded.

## BASE ALGORITHMS

This section introduces two fundamental algorithms: the Bron–Kerbosch algorithm (*Bron & Kerbosch, 1973*), the foundation for maximal clique listing, and the ARBO algorithm (*Chiba & Nishizeki, 1985*), designed explicitly for k-clique listing. Many subsequent algorithms either build upon these base algorithms or incorporate enhancements inspired by them. Thus, it is essential to present these foundational algorithms first before delving into the details of others.

The Bron–Kerbosch algorithm (*Bron & Kerbosch, 1973*) presents a seminal method for identifying all maximal cliques within an undirected graph. This algorithm employs a backtracking strategy to systematically traverse the graph's vertices and edges, rigorously enumerating all possible maximal cliques. The initial approach of the Bron–Kerbosch algorithm makes a recursive call for every clique, so this causes inefficiency, especially in graphs with many non-maximal cliques. To improve efficiency, a strategy is provided that involves strategically selecting a pivot vertex from the graph. The vertices with higher degrees are prioritized for this selection. This strategy eliminates the redundant checks by focusing the search on the pivot's neighboring vertices. Any maximal cliques among the pivot's neighbors would also be found when testing the pivot itself or its non-neighboring vertices. The vertices adjacent to the current clique are incrementally added to explore the maximum possible expansion of the clique until no more vertices can be appended. All possible maximal cliques are explored by traversing exhaustively and avoiding redundant paths. The presented Algorithm 1 (*Bron & Kerbosch, 1973*) finds all maximal cliques, including all $R$ vertices, some $P$, and none in $X$. The time complexity of this algorithm is $O(3^{n/3})$, and the space complexity is $O(m+n)$, where $n$ represents the number of vertices, and $m$ is the number of edges.

---

**Algorithm 1** BronKerboschWithPivoting($R, P, X$) (Bron and Kerbosch, 1973)

---

1: **if** $P = \{\}$ and $X = \{\}$ **then**
2:     **report** $R$ as a maximal clique
3: **end if**
4: **choose** a pivot vertex $u$ in $P \cup X$
5: **for** each vertex $v$ in $P \setminus N(u)$ **do**
6:     BRONKERBOSCHWITHPIVOTING($R \cup \{v\}, P \cap N(v), X \cap N(v)$)
7:     $P \leftarrow P \setminus \{v\}$
8:     $X \leftarrow X \cup \{v\}$
9: **end for**

---

The article **ARBO** (*Chiba & Nishizeki, 1985*) introduces graphlet counting algorithms for triangles, quadrangles, complete subgraphs, and cliques using the arboricity concept. It discusses efficient methods for computing the arboricity of a graph. The algorithm ARBO operates by selecting a vertex $v$ within the graph and scanning the edges of the subgraph induced by $v$'s neighbors to identify pattern subgraphs containing $v$. Notably, this algorithm employs an iterative search for each vertex $v$ in a non-increasing order of degree. Then, $v$ is removed after processing to prevent duplication and provide computational efficiency. Thus, it also ensures a systematic and comprehensive enumeration of subgraphs. The ARBO has a time complexity of $O(km\alpha^{k-2})$ and a space complexity of $O(m+n)$, where $\alpha$ represents the arboricity of the graph, $n$ is the number of vertices, $m$ is the number of edges, $k$ is the size of the cliques being examined.

Algorithm 2 is the pseudocode of ARBO, which starts by sorting vertices by degree order. Then, constructs and induced subgraphs from the neighbors of each vertices. The algorithm recursively searches (k-1)-cliques in the neighborhood of the current vertex

---

**Algorithm 2** The Algorithm ARBO (Chiba and Nishizeki, 1985)

1: **function** ARBO($G$, $k$)
2:   Let *Cliq* denotes ∅.
3:   Let $V$ denotes the list of degree ordered vertices in $G$.
4:   **for** each vertex $v$ **in** $V$ **do**
5:     $NbrList_v \leftarrow$ GETNEIGHBORS($G$, $v$)
6:     $G_{N_v} \leftarrow$ GETINDUCEDSUBGRAPH($G$, $NbrList_v$)
7:     $Cliq_v \leftarrow$ LISTCLIQUES($G_{NbrList_v}$, $k-1$, $\{v\}$)
8:     $Cliq \leftarrow Cliq \cup Cliq_v$
9:     DELETEVERTEX($G$, $v$)
10:  **end for**
11:  **return** *kCliques*
12: **end function**

13: **function** LISTCLIQUES($G$, $l$, $C$)
14:   **if** $l = 2$ **then**
15:     **return** $\{\{u, v\} \cup C | (u, v) \in E(G)\}$
16:   **end if**
17:   $lCliq \leftarrow ∅$
18:   **for** $u$ **in** $V(G)$ **do**
19:     $NbrList_u \leftarrow$ GETNEIGHBORS($G$, $u$)
20:     $G_{N_u} \leftarrow$ GETINDUCEDSUBGRAPH($G$, $NbrList_u$)
21:     $lCliq_u \leftarrow$ LISTCLIQUES($G_{N_u}$, $l-1$, $C \cup \{u\}$)
22:     $lCliq \leftarrow lCliq \cup lCliq_u$
23:     DELETEVERTEX($G$, $u$)
24:   **end for**
25:   **return** $lCliq$
26: **end function**

---

using the *ListCliques* function. The processed vertices are removed from the graph at the end of each iteration to eliminate the duplicate discovery of cliques in subsequent steps.

In summary, these two algorithms, Bron–Kerbosch and ARBO, can be seen as the foundation of other methods for clique enumeration. Therefore, their detailed analysis and pseudocodes are presented in this section to make the other sections more understandable. Although both algorithms serve the same purpose, they differ in their innovative strategies, efficiency, and computational characteristics. Both algorithms form the foundation of many algorithms in literature and offer critical contributions to clique counting problems with different computational advantages.

## EXACT METHODS

Exact algorithms for counting cliques fall into two main categories: those that enumerate cliques, where each is explicitly identified in the graph, and those that count cliques

without explicitly listing each. Identifying each clique is computationally intensive due to the combinatorial explosion. Counting-based algorithms determine the total number of cliques using combinatorial methods. It is optional to list all cliques, especially when there is a need for more efficient algorithms. The following subsections describe clique counting algorithms based on either enumeration or counting methodologies.

### Enumeration

In the enumeration techniques, each clique is identified explicitly. The algorithms generally employ a recursive backtracking strategy similar to the Bron–Kerbosch algorithm (*Bron & Kerbosch, 1973*). In this method, a potential clique is expanded iteratively with a pruning strategy for an option that is not promising for cliques. Identifying each clique presents comprehensive information about the graph. However, especially for large and dense datasets, the number of possible cliques grows exponentially as the graph size increases. So, working on such large and dense datasets becomes computationally costly.

The Algorithm 3 presents a pseudocode for clique enumeration based on the ARBO algorithm using degeneracy orientation. The algorithm, derived from the work of *Jain (2020)*, has been further refined to enumerate all k-cliques within a given graph G. All cliques are stored in a list during the enumeration steps and initialized at the algorithm's beginning. Then, a directed acyclic graph (DAG) is constructed using degeneracy orientation. For each vertex, the algorithm recursively finds all $(k-1)$-cliques in the out-neighborhood of the current vertex and adds the current vertex to each resulting $(k-1)$-clique, forming new $k$-cliques. At the end of the algorithm, it returns the list of cliques.

The algorithm proposed by *Akkoyunlu (1973)* is equivalent to the Bron–Kerbosch algorithm, even though they are explained differently. Both algorithms build the same search tree structure and yield the same results. The algorithm proposed by Akkoyunlu efficiently finds maximal cliques in large graphs by decomposing the problem into smaller, non-overlapping sub-problems and managing them using a stack-based approach. It begins by splitting the problem into smaller, disjoint sub-problems to avoid generating duplicate or sub-maximal cliques. Subsequently, it employs a push-down stack to store partially solved sub-problems, minimizing memory usage by focusing on the current task. The algorithm iteratively divides each sub-problem into two disjoint parts—one including a selected element and one excluding it—before pushing them onto the stack for processing. This iterative refinement continues until specific criteria are met. At this point, the algorithm applies a particular method to determine the maximal clique associated with the subset. The algorithm systematically explores the graph structure through these steps to identify all maximal cliques efficiently.

The paper **MACE** (*Makino & Uno, 2004*) comprehensively explores algorithms tailored for enumerating both maximal and bipartite cliques within graphs. For maximal cliques, it introduces two distinct strategies. The first approach utilizes matrix multiplication, capitalizing on the parent–child relationship inherent in maximal cliques to efficiently compute their children. This method constructs adjacency matrices to identify valid child cliques, resulting in a streamlined computation process that significantly improves efficiency, especially in denser graphs. The algorithm's time complexity is $O(knm\alpha^{k-2})$,

---

**Algorithm 3** BruteForceCliqueEnumeration(G, k)

1: Let $n$ denote the number of vertices in $G$.
2: Let $V$ denote the set of vertices in $G$.
3: **if** $k = 1$ **then**
4:     **return** Each vertex in $V$ as a singleton clique
5: **else if** G is a clique **then**
6:     **return** All combinations of size $k$ from $V$
7: **end if**
8: Let *Cliques* denotes Ø
9: Order the vertices of G using degeneracy ordering.
10: Convert it into a Directed Acyclic Graph (DAG) DG.
11: **for** each vertex $v \in V$ **do**
12:     Let $N_v^+ \leftarrow$ GETOUTGOINGNEIGHBORS($DG, v$)
13:     Let *subCliques* $\leftarrow$ BRUTEFORCECLIQUEENUMERATION( $(N_v^+, k-1)$)
14:     **for** each clique $C$ in *subCliques* **do**
15:         Add $v$ to $C$
16:         Add $C$ to *Cliques*
17:     **end for**
18: **end for**
19: **return** *Cliques*

---

and space complexity is $O(m+n)$, where $n$ is the number of vertices, $m$ is the number of edges, $\alpha$ is the arboricity of the graph, and $k$ is the clique size. The second algorithm for maximal cliques leverages the maximum degree of the graph. It recognizes that in sparse graphs, each maximal clique (except the lexicographically largest one) can have at most $\triangle^2$ children, in which $\triangle$ represents the maximum degree. By avoiding the explicit construction of the complete set of candidate child indices and checking candidates in lexicographic order, this method reduces computation time, particularly benefiting graphs with small maximum degrees.

The paper proposed by *Tomita, Tanaka & Takahashi (2006)* introduces a depth-first search algorithm for efficiently generating all maximal cliques in an undirected graph, leveraging pruning techniques reminiscent of the Bron–Kerbosch algorithm (*Bron & Kerbosch, 1973*). Unlike Bron–Kerbosch, which directly enumerates maximal cliques, this new algorithm outputs them in a tree-like structure, conserving memory space. It operates by iteratively expanding a global variable $Q$, representing the current clique, from an empty starting point to larger cliques. At each step, the algorithm examines the intersection of neighborhoods of vertices in $Q$, determining if it forms a maximal clique. If not, it explores potential extensions by recursively considering induced subgraphs. During the search process, the algorithm maintains two lists called FINI (processed vertices) and CAND (remaining candidates). The $Q$ is expanded only to the vertices in CAND. This minimizes the unnecessary exploration. Another strategy to reduce the number of vertices needing further exploration is to choose vertices from the neighborhood intersection. When a

maximal clique is discovered, the algorithm prints a marker instead of the clique itself. The cliques can be reconstructed from this output. The complexity of the algorithm is $O(3^{(n/3)})$, and the space complexity is $O(m+n)$, where $n$ is the number of vertices, and $m$ is the number of edges.

The algorithm (*Eppstein, Löffler & Strash, 2010*) presents a variation of the Bron–Kerbosch algorithm (*Bron & Kerbosch, 1973*). This algorithm orders the vertices according to degeneracy ordering. Then, the neighbors of each vertex are divided into two sets: $P$ and $X$. $P$ is the vertices that follow the current vertex in order of degeneration, while $X$ is the set of vertices that precede it. Thus, the size of $P$ is limited by the graph's degeneracy. This algorithm uses the Bron–Kerbosch algorithm with the parameters $P$, $X$, and current vertex. The pivot vertex is selected from the $P$ and $X$ sets during the recursive iteration. This strategy optimizes the complexity of the Bron–Kerbosch algorithm by reducing the number of recursive calls. The time complexity is $O(dn3^{d/3})$, and the space complexity is $O(m+n)$, where $n$ is the number of vertices, $m$ is the number of edges, and $d$ is the degeneracy of a graph.

The **pbitMCE** method (*Dasari, Ranjan & Mohammad, 2014*) employs a degeneracy ordering strategy similar to that presented by *Eppstein, Löffler & Strash (2010)*. However, it uses a different strategy to represent the subgraphs. The algorithm presents a data structure called a partial bit adjacency matrix (pbam). This pbam comprises sets of bit vectors. It facilitates representing the necessary information for efficient vertex processing. This algorithm is implemented on the Hadoop framework. The algorithm starts by ordering vertices and determining the degeneracy $d$ of the graph. Thus, each vertex has at most $d$ neighbors appearing later in the ordering. Subsequently, each vertex's adjacency list is partitioned into pre and post-lists, containing vertices with lower and higher degeneracy orders. This pbam consists of sets of bit vectors, each corresponding to vertices in the pre and post-lists, encoding connections between the post-list and those in the candidate set $P$. The pbam is generated using a renumbering technique to assign unique identities to vertices in $P$ and $X$. Following a similar structure to the algorithm proposed by *Eppstein, Löffler & Strash (2010)*, pbitMCE counts maximal cliques by computing the sets $P$ and $X$ for each vertex v in the degeneracy ordering and using the algorithm introduced by *Tomita, Tanaka & Takahashi (2006)* for efficient exploration of the v-rooted search tree. Each clique is associated with the node with the lowest number. This facilitates the unique reporting of each maximal clique. The complexity is $O(kn3^{k/3})$, where the k-degree of a graph is defined as the minimum value such that every vertex $v$ has at most $k$ neighbors with a degree greater than or equal to the degree of $v$. It is hard to compute the degeneracy ordering of vertices in a distributed environment. So, implementing pbitMCE on Hadoop is a challenging task. This task requires extensive inter-node communication. To reduce this complexity, the paper proposes to explore alternative vertex orderings, such as degree-based ordering in some scenarios.

The **kClist** (*Danisch, Balalau & Sozio, 2018*) algorithm improves the ARBO algorithm (*Chiba & Nishizeki, 1985*) for listing all k-cliques. The degeneracy orientation is used, and a directed acyclic graph (DAG) is constructed to eliminate duplicate discovery of cliques. Besides, the algorithm utilizes parallelization techniques and special data structures to

improve its performance for large-scale graph analysis. The time complexity of kClist is $O(m\alpha^{k-2})$, where $\alpha$ represents the arboricity of the graph $G$, $m$ is the number of edges, $k$ is the size of the cliques being examined. The algorithm has the $O(m + P\alpha^2)$ space on $P$ processors when using a work-stealing scheduler (*Blumofe & Leiserson, 1993*; *Shi, Dhulipala & Shun, 2021*).

A new heuristic for k-clique listing and counting algorithm, using a color ordering method derived from greedy graph coloring techniques (*Hasenplaugh et al., 2014*; *Yuan et al., 2017*) are proposed by *Li et al. (2020)*. A graph colorization technique using a greedy coloring algorithm is employed to graph, and distinct color values are assigned to adjacent nodes from 1 to the chromatic number $x$. The nodes are sorted in descending order based on this color number, and then a DAG is constructed. Thus, inefficient search paths are eliminated during the iterative enumeration process. The time complexity of $O(km(\frac{\Delta}{2})^{k-2})$, and the space complexity is $O(m + n)$, where $k$ is the clique size, $\Delta$ is the maximum degree, and $m$ is the number of edges, $n$ is the number of vertices. Additionally, this paper provides a thorough experimental evaluation of existing algorithms for listing and counting k-cliques.

*Yuan et al. (2022)* presents two k-clique listing algorithms: **SDegree** and **BitCol**. These algorithms aim to accelerate the k-clique listing algorithms with merge-based set intersections and parallelism. For this purpose, the paper proposes two pre-processing techniques: Pre-Core and Pre-List. First, Pre-Core reduces the search space by removing redundant vertices not contained in a k-clique. Then, Pre-List checks all connected components; if a connected component is a clique, it directly lists and removes cliques. The SDegree algorithm employs degree-based orientation and constructs a DAG. The novelty of this algorithm is to use the merge-join strategy while merging two vertex sets rather than the hash join, which is used by the other algorithms in the literature. If the vertex sets are ordered, the merge-join strategy efficiently merges these sets. The BitCol algorithm improves the SDegree algorithm by employing degeneracy and color-based ordering techniques and compressing the vertex sets using bitmaps. First, the input graph is converted to DAG using degeneracy orientation; then, the algorithm iteratively searches each vertex neighborhood. An induced subgraph is obtained from the current node's neighborhood, and the DAG of this subgraph using color-based orientation is constructed. This algorithm also uses advanced parallelization strategies for efficient k-clique listing. The time complexity of both algorithm is $O(km(\frac{\Delta}{2})^{k-2})$. The space complexity of SDegree is $O(m + kN\Delta)$ and BitCol is $O(m + N\frac{\Delta^2}{L})$, where $k$ is the clique size, $N$ is the number of threads, $\Delta$ is the maximum out-degree, $L$ is the size of nodes that each number can represent, and $m$ is the number of edges.

The k-clique listing algorithms traditionally employ a vertex-based branching strategy, where larger cliques are constructed incrementally by adding a single vertex to an existing clique. A new algorithm **EBBkC** (*Wang, Yu & Long, 2024*) in the literature proposes an edge-based branching strategy. Instead of adding a single vertex, it tries to obtain larger cliques more efficiently by adding two nodes with edges between them, thus narrowing the search space. The algorithm incorporates three distinct edge-ordering methods to optimize the branching process. The first is truss-based edge ordering, which leverages

truss decomposition to order edges to minimize the size of the resulting subgraphs, thus enhancing efficiency. The second is color-based edge ordering, which utilizes vertex colorings to prune branches, effectively reducing the number of candidate cliques and further improving performance. The third method is a hybrid approach, combining the strengths of truss-based and color-based ordering to provide theoretical and practical improvements. Besides, the paper introduces a method to terminate branches early if the subgraph is a dense structure like a clique or each vertex is connected to at least $k - 2$ other vertices within the subgraph, leveraging efficient combinatorial algorithms to list cliques in these cases to increase efficiency. This algorithm has the $O(md + k \cdot m \cdot \left(\frac{\tau}{2}\right)^{k-2})$ time complexity, where $d$ is the degeneracy of the graph, $\tau$ is the maximum truss number of the graph (*Wang, Yu & Long, 2024*), $k$ is the clique size, $m$ is the number of edges. This algorithm presents better time complexity than the vertex-based branching algorithms (*Chiba & Nishizeki, 1985*; *Makino & Uno, 2004*; *Danisch, Balalau & Sozio, 2018*), which have $O(km\alpha^{k-2})$. The authors demonstrate that the $\tau$ is smaller than $d$, leading to better performance in edge-based branching. The space complexity is $O(m + n)$. This paper also incorporates parallelism techniques to further enhance the efficiency of the proposed algorithm and provides a comparison with the state-of-the-art k-clique listing algorithms depending on vertex-based branching strategy.

We review various enumeration-based k-clique algorithms, focusing on their methodology and complexity. Bron–Kerbosch (*Bron & Kerbosch, 1973*) and Akkoyunlu (*Akkoyunlu, 1973*) are similar algorithms because they construct similar search trees but are presented in different terms. They utilize recursive backtracking and stack-based approaches to identify maximal cliques. A matrix multiplication technique and the information of maximum degree in the graph are used for efficient clique enumeration in the MACE algorithm (*Makino & Uno, 2004*). *Tomita, Tanaka & Takahashi (2006)* proposes an improvement based on a depth-first search approach to Bron–Kerbosch, and they use pruning techniques for optimal memory usage. *Eppstein, Löffler & Strash (2010)*, and pbitMCE (*Dasari, Ranjan & Mohammad, 2014*) algorithms leverage degeneracy ordering and parallel processing for the performance in large-scale graph analysis based on Bron–Kerbosch algorithm. The kClist algorithm also (*Danisch, Balalau & Sozio, 2018*) provides similar contributions based on the ARBO algorithm. A heuristic method by *Li et al. (2020)* uses color order to optimize the search process by pruning unproductive paths. The paper (*Yuan et al., 2022*) presents two parallel k-clique listing algorithms, SDegree and BitCol, which use merge-based set intersections and preprocessing techniques to provide a time and space-efficient approach than the algorithms proposed by *Li et al. (2020)*. The EBBkC (*Wang, Yu & Long, 2024*) introduces a new edge-based branching strategy and edge-based ordering techniques to improve ARBO and presents better time complexity.

## Counting

Counting-based algorithms determine the total number of cliques in a graph without needing to identify and list each clique explicitly. Combinatorial methods, dynamic programming, or algebraic approaches are used to calculate the overall count of cliques. The *Ahmed et al. (2015)* algorithm and the ESCAPE (*Pinar, Seshadhri & Vishal, 2017*)

algorithm provide some combinatorial methods for graphlet types, including cliques of different sizes, but they are beyond the scope of our paper. Identifying each clique is a challenging task. For this reason, the literature diverges its direction counting-based algorithms to provide more scalable and efficient approaches for larger datasets.

Algorithm 4, provided by *Jain (2020)*, presents k-clique counting algorithms based on ARBO (*Chiba & Nishizeki, 1985*). This algorithm presents a modified version, rearranged for counting rather than listing (*Jain, 2020*), leverages degeneracy ordering to partition the graph into multiple subgraphs, and then recursively counts cliques within these subgraphs. It begins by initializing variables to store the number of vertices $n$ and the set of vertices $V$. According to this algorithm, if k = 1, the number of vertices $n$ is returned as each vertex forms a singleton clique. If the graph is a clique, the algorithm returns the number of k-combination of $n$ vertices. After that, a DAG is constructed using degeneracy orientation. The algorithm searches the (k-1)-cliques in the outgoing neighborhood of each vertex. Finally, it returns the number of k-cliques at the end of the procedure.

---

**Algorithm 4** BruteForceCliqueCounting(G, k) (Jain, 2020)

---

1: Let $n$ denotes the number of vertices in $G$.
2: Let $V$ denotes the set of vertices in $G$.
3: **if** $k = 1$ **then**
4:     **return** $n$
5: **else if** G is a clique **then**
6:     **return** $\binom{n}{k}$
7: **end if**
8: Let $C_k = 0$
9: Order the vertices of G using degeneracy ordering.
10: Convert it into a Directed Acyclic Graph (DAG) DG.
11: **for** each vertex $v \in V$ **do**
12:     Let $N_v^+ \leftarrow$ GETOUTGOINGNEIGHBORS($DG, v$)
13:     $C_k = C_k + \text{BruteForceCliqueCounting}(N_v^+, k-1)$
14: **end for**
15: **return** $C_k$

---

The paper proposed by *Finocchi, Finocchi & Fusco (2015)* presents two exact and approximate solutions for the issue of counting the number of k-cliques in large-scale graphs by focusing on theoretical and experimental aspects. It introduces parallel solutions using the degree orientation technique in the MapReduce framework. First, it provides an exact approach, then presents a sampling-based approach that significantly reduces the exact approach's computational demands. The second step identifies all the other nodes with a lower degree than those nodes and forms a triangle with them. In the third and final stage, the reduce phase, the algorithm finds clique patterns for each node using the information collected in the previous round. The paper explores approximate counting using two sampling strategy variants and the exact counting approach. The exact algorithm of this study is efficient for counting up to seven-node cliques in relatively small datasets;

approximate counts are provided for larger datasets due to computational complexity. This algorithm requires $O(m^{k/2})$ computational effort and $O(m+n)$ space, where $m$ represents the number of edges in the graph, $n$ is the number of vertices, and $k$ is the size of the examined cliques.

The **Pivoter** is designed by *Jain & Seshadhri (2020a)* to deal with the challenge of exact counting k-cliques in graphs, especially as the size of $k$ increases. Pivoter utilizes pivoting to construct a Succinct Clique Tree (SCT), which provides a compressed representation of all cliques in the graph. SCT provides a strategy different from existing methods that explicitly enumerate every clique. A vertex $v$ is selected to construct SCT, and the neighborhood of $v$ is explored to form larger cliques recursively. The aim is to find all cliques that include $v$. The neighbors of $v$ form the candidate set, and a pivot vertex is chosen from the candidate set. A vertex that maximizes the number of neighbors it shares with other vertices in the candidate set is chosen as a pivot. The search space is divided into two parts: cliques containing the pivot vertex and cliques not containing the pivot vertex. The splitting search space helps in reducing the number of recursive calls. After selecting the pivot, the algorithm recursively searches the remaining vertices in the candidate set to form cliques. This manner is repeated for each vertex in the graph. Using SCT, Pivoter counts k-cliques cliques of any size without complete enumeration and reduces the recursion tree of backtracking algorithms. Thus, Pivoter states it overcomes scalability issues and achieves accurate clique counts in large graphs. Key contributions are the counting cliques for both globally and locally, for each vertices and edges, and the creation of SCTs through pivoting. Besides, a parallel version is also presented to enhance the algorithms' performance and scalability on large datasets. The Pivoter has $O(n\alpha 3^{\alpha/3})$ time complexity, and $O(m+n)$ space complexity, where $n$ represents the number of vertices, $m$ is the number of edges, and $\alpha$ is the degeneracy of the graph (*Jain & Seshadhri, 2020a*). However, as stated in the research paper, even the parallel version of the algorithm has limitations on large graphs. Pivoter needs help to compute clique counts beyond $k = 10$.

As a summary, *Finocchi, Finocchi & Fusco (2015)* proposes two parallel exact and approximate algorithms that handle the limitation of the exact approach based on the MapReduce framework. The Pivoter brings innovation using an SCT data structure that eliminates complete enumeration. Both algorithms offer promising solutions for large-scale graph analysis.

## APPROXIMATE METHODS

Since the exact approaches are challenging for massive datasets, the approximate solutions have significantly attracted attention in the literature and aim to estimate the number of k-cliques in a graph dataset instead of thoroughly examining every possible combination. Typically, sampling strategies and statistical methods are employed to estimate clique counts close to the exact value. The sampling strategies select a subset of nodes and edges from a graph. These subsets are expected to capture the essential structural properties of a graph while staying computationally manageable. For this purpose, a suitable sampling size must be selected. The sampling size affects both the computational efficiency and accuracy

of the algorithm. The sampling strategies represent the large datasets by a smaller subset and enable the analysis of large datasets that are impractical to handle using exact algorithms. A well-chosen sample size accelerates analysis, conserves computational resources, and provides meaningful insights.

In this context, we explore various sampling strategies from the literature and focus on those specifically used in clique counting algorithms.

### Random sampling

In the random sampling method, vertices or edges are uniformly selected regardless of their attributes or relevance within the graph. This method provides an unbiased estimation and graph representation but risks the critical structural components or nodes crucial to graph dynamics that must be paid attention to.

The **Turán-shadow** (*Jain & Seshadri, 2017*) algorithm is a randomized approach that aims to estimate the number of $k$-cliques (where $k \leq 10$) in a graph based on Turán's theorem (*Turán, 1941*). Turán's theorem provides insights about a graph's maximum number of edges without having a $(k+1)$-clique according to the number of vertices. Formally, Turán's theorem states that in a graph $G = (V, E)$ with $n$ vertices that do not contain a clique of size $k+1$ (where $k$ is greater than zero), the number of edges is bounded by $\left(1 - \frac{1}{k}\right)\frac{n^2}{2}$. That means if the edge density of a graph exceeds $\left(1 - \frac{1}{k-1}\right)$, it must contain a $k$-clique. The algorithm starts by orienting the graph according to degeneracy ordering to reduce the search space and then continues creating the TuránShadow. It explores the neighborhoods of vertices iteratively to identify denser subgraphs. If the edge density of an out neighborhood exceeds the Turán density threshold for $(k-1)$-cliques, the induced subgraph is added to the TuránShadow. Otherwise, the process is applied recursively to find denser sets. The resulting TuránShadow comprises sets with densities above the Turán threshold, forming a collection of potential $k$-cliques. A sampling strategy is employed to randomly select subsets of vertices from these sets, which are then checked for clique formation. The time complexity is $O(n\alpha^{k-1})$, and the space complexity is $O(n\alpha^{k-2} + m)$, where $n$ is the number of vertices, $m$ is the number of edges, $\alpha$ is the degeneracy, and $k$ is the clique size.

This **YACC** algorithm (*Jain & Tong, 2022*) is an extension of the Turán-shadow algorithm (*Jain & Seshadri, 2017*) to improve the counting of large cliques in graphs. Algorithms Turán-shadow and Pivoter (*Jain & Seshadri, 2020a*) previously proposed algorithm by the authors excel at counting small cliques, but they face challenges with larger cliques. YACC improves approximate clique counting by reducing the recursion tree's size and exploiting insights from real-world graph structures. YACC relaxes the stopping condition of Turán-shadow, which relies on a fixed density threshold from Turán's theorem. Thus, YACC efficiently identifies dense subgraphs with a more adaptable stopping condition by significantly reducing the size of the recursion tree and the computation time. This improvement makes clique counting possible for challenging graphs like com-lj (*Leskovec & Krevl, 2014*). The framework enhances control over the construction-sampling balance by introducing a parameter, μ, which affects the balance between computational complexity and accuracy of results. This parameter redefines what is needed for a graph

to be considered dense. The algorithm divides the graph into two regions: dense and sparse. A sampling strategy is employed to estimate the clique counts in dense regions. The cliques in the sparse area counted exactly in a recursive manner. As a result, the efficiency and adaptability of Turán-shadow are improved by YACC with heuristics in practical applications. The complexity of time and space is similar to TuránShadow.

The Turán-shadow, YACC (extended version of Turán-shadow), and DP-ColorPath (*Ye et al., 2024*), which will be explained in the color-based sampling section, are outstanding approximate k-clique counting algorithms based on a sampling strategy. A typical step of these algorithms is constructing a sampling space consisting of dense subgraphs containing k-cliques. These algorithms then sample a fixed element from the sampling space to estimate k-clique counts. However, this fixed sampling does not guarantee accuracy. The **SR-kCCE** (*Chang, Gamage & Yu, 2024*) algorithm presents a sampling-stopping strategy that guarantees accuracy while providing efficiency. Like the Turán-shadow, YACC, and DP-ColorPath algorithms, this algorithm consists of two steps: constructing the sampling space and sampling randomly from that space to estimate k-clique counts. The algorithm becomes inefficient when the sampling space construction time is high, especially for large datasets. On the other hand, if the sampling space is not refined from the non-cliques, the relative error worsens. The SR-kCCE algorithm constructs a balance between these two steps. This algorithm estimates the expected duration of the sampling phase. When this expected duration is approximately the same as the construction/refinement sampling space, called a shadow, the algorithm stops the refinement sampling space. Thus, it ensures that both phases are balanced regarding computational effort. This algorithm improves k-clique estimation compared to previous methods, especially for large datasets.

## Rejection sampling

This method is a Monte Carlo-based (*Mackay, 1998*) technique that generates samples from a target probability distribution function when direct sampling is impractical or infeasible. Firstly, a simpler distribution, which is relatively easy to sample from and covers the support of the target distribution, also known as the proposal distribution, is chosen. The samples are generated from this distribution, and then a decision is made whether to accept or reject them based on their adherence to the characteristics of the target distribution. At each proposed sample point, the ratio of the probability density function of the target distribution is calculated to that of the proposal distribution. Accordingly, it is decided whether the sample is accepted or rejected. The purpose of this selective approach is to ensure that the samples generated are consistent with the properties of the target distribution. The effectiveness of rejection sampling depends on choosing an appropriate proposal distribution. Overly simplistic or complex choices can lead to inefficiency.

*Eden et al. (2017)* introduce a sublinear-time algorithm for triangle counting that defies the conventional linear-time norm for such computations. The paper uses degree, neighbor, and pair queries within the standard query model for sublinear algorithms on general graphs. Building on this significant advancement, an algorithm called **ERS** (*Eden, Ron & Seshadhri, 2018*) is introduced to extend its application beyond triangle counting. It aims to approximate the count of k-cliques within sublinear time, thus covering the

previously established result for $k = 3$. The algorithm selects a sample set of vertices to estimate the number of k-cliques connected to this subset. A crucial aspect is to sample each k-clique connected to the sample set with almost equal probability. However, random sampling can compromise the selection of high-degree vertices likely to form cliques. The algorithm randomly samples high-degree vertices and tries to strike a careful balance between predicting cliques formed by high-degree vertices and cliques formed by low-degree vertices. A uniform edge $(u, v)$ is sampled, and more vertices are added to that edge iteratively to attempt to form a k-clique. Depending on whether $v$ is a low- or high-order vertex, the algorithm employs different sampling strategies, including uniform neighbor selection and rejection sampling. The complexity of algorithm $\tilde{O}\left( \frac{n}{C_k^{1/k}} + \frac{m^{k/2}}{C_k} \right)$, where $n$ is the number of vertices, $C_k$ is the number of $k$-cliques, and $m$ is the number of edges. The algorithm has the $O(m + n)$ space complexity.

## Color-based sampling

This sampling method samples each edge in the graph with a probability $p$, where $N = 1/p$ is an integer. Each vertex is randomly assigned one of $N$ colors with equal probability $p$. An edge is designated as monochromatic if both endpoints share the same color. Then, a subgraph is constructed from these monochromatic edges. Initially, this method is proposed by *Pagh & Tsourakakis (2012)* and counts the number of triangles (three-cliques) in this subgraph using either an exact or approximate triangle counting method. The resulting estimation value is scaled by multiplying $p^{-2}$ for the total number of triangles in the original. The focus is to create a subgraph that preserves the original graph's structural attribute based on the connectivity of vertices according to their color. Thus, large graphs can be analyzed computationally more efficiently using smaller samples representing them. This can also highlight the more meaningful structural patterns or cliques by eliminating less significant edges in the graph. *Shi, Dhulipala & Shun (2021)* proposes an extension of this approach for the applicability of larger clique sizes.

The paper DP-ColorPath proposed by *Ye et al. (2024)* combines exact and approximate solutions to enable working with large and dense datasets. The graph is partitioned into dense and sparse regions. The algorithms that implement exact clique counting use the efficiency of the Pivoter algorithm for sparse areas and adapted sampling-based techniques for denser areas. Initially, a linear-time greedy coloring process is employed (*Hasenplaugh et al., 2014*; *Yuan et al., 2017*), establishing a non-decreasing node ordering based on color values and constructing a directed acyclic graph (DAG) accordingly. Following the computation of a DAG, nodes are partitioned into sparse and dense regions based on the average degree of the neighborhood subgraph. The Pivoter algorithm accurately computes (k-1)-clique counts in the sparse areas. At the same time, dense regions are addressed using three sampling-based methods: k-color set sampling, k-color path sampling, and k-triangle path sampling. These three algorithms utilize dynamic programming techniques and conduct uniform sampling. In the k-color set sampling method, k-color sets are selected, each consisting of k nodes with unique colors. The k-color path sampling samples from connected k-color sets are called k-color paths. It ensures the induced subgraph

by the $k$ nodes stays connected. The most effective of the trio, k-triangle path sampling, selects connected k-color sets where any three consecutive nodes form a triangle, known as k-triangle paths. The time complexity of k-color set sampling is $O(\chi^k)$, and the space complexity is $O(m+n+\chi^k)$. The time complexity of k-color path sampling is $O(\chi^{nk}+m)$, and the space complexity is $O(kn+m)$. The time complexity of k-triangle path sampling is $O(k\Delta)$, and the space complexity is $O(km)$. The $\chi$ is the number of colors of the graph G obtained by the greedy coloring algorithm (*Hasenplaugh et al., 2014*; *Yuan et al., 2017*), $k$ is the clique size, $n$ is the number of vertices, $m$ is the number of edges and $\Delta$ is the number of triangles of the input graph.

Besides these sampling strategies, there is an algorithm BDAC (*Çalmaz & Bostanoğlu, 2024*) that presents an approach to approximate k-clique counts, especially for large datasets, without using any sampling strategy. Rather than providing an approximate result, it provides lower and upper boundary based on extremal graph theorems. To the best of our knowledge, this algorithm is the first to provide such a boundary for k-cliques. Existing methods have efficiency issues, especially beyond $k = 10$, or require accuracy and resource consumption trade-offs. The BDAC algorithm aims to overcome these challenges by using triangle density information and established extremal graph theorems such as the Turán (*Turán, 1941*), Zykov (*Zykov, 1949*), Kruskal-Katona (*Kruskal, 1963*; *Katona, 1987*), and Reiher's (*Reiher, 2016*) theorems to provide both lower and upper bounds on the k-clique count at the local (per vertex) and global levels. The Turán-shadow algorithm inspires this algorithm. It follows the vertex iterative manner as the Turán-shadow algorithm. The BDAC eliminates the construction of TuránShadow, which requires a recursion tree and the sampling phase. This paper offers consistent complexity regardless of k, making it suitable for large datasets, and demonstrates its effectiveness for k-clique counts up to $k = 50$. This algorithm, however, faces limitations with large and sparse graphs. When the density of a sparse subgraph falls below the Turán threshold, it fails to provide a minimum clique count, and the gap between the lower and upper bounds increases. The time complexity of BDAC is $O(\alpha^2)$, and the space complexity is $O(m+n+\alpha)$, where $m$ is the number of edges, $n$ is the number of vertices, $\alpha$ is the arboricity.

In conclusion, the Turán-shadow algorithm (*Jain & Seshadri, 2017*) and its extension, YACC (*Jain & Tong, 2022*), propose randomized sampling-based solutions based on Turán's theorem (*Turán, 1941*) for the k-clique counting problem. While Turán-shadow is suitable for $k$ values less than 10 and relatively small datasets, the YACC algorithm can handle $k$ values up to 40 and has shown results for large datasets that were not previously reported in the literature. The algorithm ERS presents the sublinear-time solution (*Eden, Ron & Seshadri, 2018*) defying traditional linear-time norms. The algorithms proposed by DP-ColorPath (*Ye et al., 2024*) combine exact and sampling-based techniques to handle large and dense graphs. The fixed number of samples used by algorithms like Turán-shadow, YACC, and DP-ColorPath impacts their accuracy; the SR-kCCE (*Wang, Yu & Long, 2024*) algorithm addresses this limitation by balancing the construction sampling space and sampling phases. The algorithm both provides efficiency for k-clique estimation and guarantees accuracy. Unlike these approximation algorithms, the BDAC algorithm

provides a boundary instead of an estimation for the k-clique count without relying on any sampling strategy or recursive process.

## PARALLELIZATION STRATEGIES

Parallelization strategies comprise methodologies designed to break down intricate computational tasks into smaller, manageable components, executed concurrently to bolster efficiency and tackle scalability challenges. Leveraging shared memory systems, MapReduce frameworks (*Dean & Ghemawat, 2004*), and distributed platforms like Hadoop harness computational resources. These methods decompose computations into parallel sub-tasks executed simultaneously across distributed systems. Significant performance enhancements are achieved by distributing workload and executing tasks concurrently, circumventing the constraints of sequential processing and effectively addressing computationally intensive problems. Multiple parallelization strategies exist, but this work focuses on the methods used in clique counting.

### Shared memory

It is a programming model where numerous processes or threads can access and modify a shared memory space, enabling efficient communication and data sharing without explicit message passing. All processes/threads have access to the same address space in shared memory systems, allowing them to read from and write to shared variables for synchronization. In shared memory systems, load imbalance is a critical issue, especially for clique counting. This is because different parts of the graph have different levels of complexity, with some subgraphs containing many cliques and others very few.

*Shi, Dhulipala & Shun (2021)* introduces a series of parallel algorithms designed to address challenges in k-clique counting and densest subgraph detection. At its core, the ARB-Count algorithm enhances Chiba-Nishizeki's approach (*Chiba & Nishizeki, 1985*) by leveraging low out-degree orientations of graphs, achieved through efficient parallel implementations of algorithms such as those by *Goodrich & Pszona (2011)* and *Barenboim & Elkin (2008)*. This orientation technique aims to direct the edges of a graph to minimize the number of outgoing edges (out-degrees) from each vertex. Doing so simplifies the recursive clique counting process by limiting the number of vertices that must be considered at each step. This directly improves the algorithm's time and space complexity. The orientation reduces overall work by peeling vertices in parallel, resulting in a poly-logarithmic span and more efficient performance. ARB-Count exploits parallelism by recursively intersecting out-neighbors of vertices to build k-cliques efficiently. Utilizing parallel hash tables, filtering, and reduction operations, it achieves notable speed-ups, particularly for large graphs and values of k. The complexity of ARB-Count is $O(m\alpha^{k-2})$. Additionally, the paper presents ARB-PEEL and ARB-APPROX-PEEL algorithms for approximating k-clique densest subgraphs, which capitalize on parallel k-clique counting to peel vertices in parallel based on their k-clique counts iteratively. Colorful sparsification technique is employed to estimate k-cliques by drawing inspiration from earlier work on approximating triangle and butterfly(bi-clique) counts (*Pagh & Tsourakakis, 2012*; *Sanei-Mehri, Sariyuce & Tirthapura, 2018*). It leverages the proposed ARB-Count algorithm as

a subroutine to achieve this approximation. The time complexity of the approximate algorithm is $O(pm\alpha^{k-2} + m)$, where $m$ is the number of edges $p = 1/c$, and c is the number of colors used. The ARB-Count algorithm requires $O(m + P\alpha)$ space on P processors.

The kClist (*Danisch, Balalau & Sozio, 2018*; *Li et al., 2020*), Pivoter (*Jain & Seshadhri, 2020a*; *Yuan et al., 2022*; *Ye et al., 2024*), and EBBkC (*Wang, Yu & Long, 2024*) algorithms similarly provide parallel solutions for clique counting, leveraging shared memory architectures.

## MapReduce

MapReduce (*Dean & Ghemawat, 2004*) is a programming model where large datasets are processed in parallel across a distributed cluster. It divides the computational task into a Map phase, where input data is split and processed in parallel, and a Reduce phase, where the intermediate results are combined and aggregated. **Hadoop** is an open-source framework that implements the MapReduce model, consisting of the Hadoop Distributed File System (HDFS) for distributed storage and the Hadoop MapReduce framework for distributed processing, handling task scheduling, data partitioning, and fault tolerance. The distributed environment presents significant bottlenecks, particularly in the computation of degeneracy ordering, which necessitates extensive inter-node communication (*Dasari, Ranjan & Mohammad, 2014*).

The algorithms pbitMCE (*Dasari, Ranjan & Mohammad, 2014*; *Finocchi, Finocchi & Fusco, 2015*) algorithms utilize the MapReduce framework for parallel implementation.

## Graphics processing units

They are specialized hardware devices optimized for parallel processing. With thousands of processing cores, graphic processing units (GPUs) are particularly adept at executing numerous computations simultaneously. By offloading computations from the CPU to the GPU, parallel tasks can be processed more efficiently. This parallel processing capability enables significant performance improvements, especially for tasks suitable for parallel execution. GPUs distribute workloads across multiple cores, allowing for concurrent execution of independent sub-tasks, which leads to enhanced efficiency and faster processing times. The GPUs have many bottlenecks explained in *Almasri et al. (2022)*, especially regarding clique counting. They require fine-grained parallelism as the computational resources are organized hierarchically. Because of this structure, it is challenging to balance the workload. The recursive nature of clique counting exacerbates this imbalance by introducing irregular workloads. In addition, GPUs have limited memory, which constrains the number of threads that can run in parallel, as each thread requires memory to store its state while traversing the search space.

*Almasri et al. (2022)* integrates the graph orientation and pivoting (*Jain & Seshadhri, 2020a*) techniques to GPU accelerate existing algorithms for counting k-cliques in graphs. These algorithms are based on vertex-centric and edge-centric parallelization strategies, with binary coding and sub-warp partitioning methods that optimize memory usage and maximize parallel resources. One process that requires the most effort in clique counting algorithms is intersection operations. If we give an example of intersection operations on

a triangle, which is the simplest of cliques (three-clique), to find the triangle formed by an edge, the intersection of the neighbors of the two nodes forming the edge is needed. This paper uses binary encoding to facilitate the intersection process and represents each vertex's induced subgraph with binary encoding. This strategy facilitates the intersection processes with bit-wise operations. On the parallelization side, sub-warp partitioning divides thread blocks into smaller groups, allowing tasks to be executed more efficiently on the GPU and helping to increase the level of parallelism. Additionally, it facilitates operations like list intersections. A hybrid version of degree and degeneracy orientation techniques is employed while orienting the graph. A comparison of vertex-centric and edge-centric parallelization strategies is provided regarding load balancing and weighing of parallelism granularity trade-offs. This paper also provides solutions for GPU memory constraints by using memory management techniques like binary encoding, pre-allocating memory for the largest potential-induced subgraph size, and substituting recursive tree traversal with an iterative method using a shared stack. The algorithm requires $O(d_{max}^2)$ space to store binary-encoded adjacency lists of induced subgraphs, where the largest induced subgraph has at most $d_{max}$ vertices.

To the best of our knowledge, ARB-Count (*Shi, Dhulipala & Shun, 2021*) provides the most efficient CPU-based parallel k-clique counting algorithm, while *Almasri et al. (2022)* offers a GPU-based parallel k-clique counting algorithm, and both algorithms provide the most efficient parallel versions of the base algorithms in the literature. The typical initial step of these algorithms is graph orientation. While *Shi, Dhulipala & Shun (2021)* employs degeneracy orientation and parallel hash tables as a parallel strategy, *Almasri et al. (2022)* introduces a hybrid version of degree and degeneracy orientation and provides GPU acceleration.

## DISCUSSION OF ALGORITHMS

This section compares the algorithms according to results reported in related research papers. Algorithms are categorized according to the main features of their approach. Table 2 displays this categorization, and each column entry provides detailed information. The column *Approximate* indicates whether an algorithm uses an approximation technique, typically sampling strategies, and specifies the type of sampling strategy if available. Otherwise, there is no entry. A similar approach is employed to *Exact* column. An algorithm may provide both approximate and exact algorithms; both exact and approximate columns indicate the corresponding strategies. The *Parallelization* column indicates whether the algorithm supports parallel execution; if the corresponding entry for the algorithm is empty, it does not support parallelization. If an algorithm is based on one of the two base algorithms mentioned in this publication and provides suggestions for improving this algorithm, the *Base algorithm* column specifies this base algorithm. Most clique counting algorithms use an orientation technique such as degree, degeneracy, or color-based method as a pre-processing step to eliminate the duplicate exploration of cliques. The undirected input graph is converted to a directed acyclic graph (DAG) using one of these orientation techniques. The column *Orientation* indicates the orientation methods used by

the algorithm; an empty entry indicates that no orientation techniques are used. The column *Objective* details the specific goal of each algorithm, indicating whether the algorithm is designed to enumerate maximal cliques or count k-cliques. Maximal and k-clique counting tasks overlap since a maximal clique can contain several smaller k-cliques. While counting maximal cliques, one can indirectly gather information about k-cliques. Many algorithms designed for k-clique counting have been inspired by maximal clique counting algorithms, particularly the Bron–Kerbosch algorithm. Several k-clique counting algorithms built upon innovations introduced by maximal clique counting techniques and improvements to *Bron & Kerbosch (1973)* have been adapted for k-clique counting. For this reason, Table 2 also covers both maximal and k-clique counting algorithms, which are differentiated from the *Objective* column. The *Time Complexity* column specifies the computational complexity of the algorithms, while the *Space Complexity* column outlines the memory requirements if such information is available; otherwise, the entry states "not reported." The explanations of complexity parameters are also included in the section detailing the algorithm.

The Bron–Kerbosch algorithm (*Bron & Kerbosch, 1973*) and ARBO (*Chiba & Nishizeki, 1985*) represent distinct approaches to the problem of enumerating cliques in a graph. Bron–Kerbosch uses a backtracking strategy, leveraging pivot vertices to thoroughly identify all maximal cliques. In contrast, ARBO relies on arboricity, scanning subgraphs induced by vertices to a decreasing degree to enumerate cliques efficiently. While the Bron–Kerbosch algorithm guarantees exhaustive coverage of all maximal cliques, the time complexity of ARBO is linked to the arboricity of the graph, making it particularly efficient for real-world graphs with low arboricity.

Akkoyunlu's algorithm (*Akkoyunlu, 1973*), although described differently, essentially mirrors Bron–Kerbosch by generating an identical search tree (*Johnston, 1976*).

The algorithms proposed by *Tomita, Tanaka & Takahashi (2006)*; *Eppstein, Löffler & Strash (2010)*; *Dasari, Ranjan & Mohammad (2014)* (pbitMCE) each present unique approaches to finding all maximal cliques in an undirected graph, sharing a common heritage rooted in the Bron–Kerbosch algorithm (*Bron & Kerbosch, 1973*). *Tomita, Tanaka & Takahashi (2006)*'s algorithm is notable for using depth-first search combined with effective pruning techniques. While these techniques are reminiscent of the Bron–Kerbosch method, *Tomita, Tanaka & Takahashi (2006)* algorithm structures the output in a memory-efficient tree-like format, unlike Bron–Kerbosch, which directly enumerates cliques without such organization.

If we compare the algorithms in terms of time and space complexity, the classic algorithms Bron–Kerbosch (*Bron & Kerbosch, 1973*; *Akkoyunlu, 1973*) have exponential time complexity due to the combinatorial nature of the problem and require linear space mainly to store recursion data. They are suitable for small to moderate-sized graphs where exact enumeration of maximal cliques is feasible. The algorithm proposed by *Tomita, Tanaka & Takahashi (2006)* has a similar complexity to Bron–Kerbosch. However, it has the advantage of slightly improved performance due to incorporating pivoting techniques. These algorithms are not suitable for large datasets due to exponential time complexity.

*Eppstein, Löffler & Strash (2010)* introduced a significant variation by incorporating degeneracy ordering, which optimizes vertex processing. This strategic ordering ensures

**Table 2  A comparison of algorithms based on different characteristics.**

| Algorithm | Approximate | Exact | Parallelization | Base algorithm | Orientation | Objective | Time complexity | Space Complexity |
|---|---|---|---|---|---|---|---|---|
| Bron–Kerbosch (*Bron & Kerbosch, 1973*) | | enumeration | | | | maximal | $O(3^{n/3})$ | $O(m+n)$ |
| (*Akkoyunlu, 1973*) | | enumeration | | | | maximal | $O(3^{n/3})$ | $O(m+n)$ |
| ARBO (*Chiba & Nishizeki, 1985*) | | enumeration | | | | k-clique | $O(km\alpha^{k-2})$ | $O(m+n)$ |
| MACE (*Makino & Uno, 2004*) | | enumeration | | | | maximal | $O(knm\alpha^{k-2})$ | $O(m+n)$ |
| (*Tomita, Tanaka & Takahashi, 2006*) | | enumeration | | Bron–Kerbosch | | maximal | $O(3^{n/3})$ | $O(m+n)$ |
| (*Eppstein, Löffler & Strash, 2010*) | | enumeration | | Bron–Kerbosch | degeneracy | maximal | $O(dn3^{d/3})$ | $O(m+n)$ |
| pbitMCE (*Dasari, Ranjan & Mohammad, 2014*) | | enumeration | MapReduce | Bron–Kerbosch | degree/degeneracy | maximal | $O(kn3^{k/3})$ | not reported |
| (*Finocchi, Finocchi & Fusco, 2015*) | color-based | enumeration | MapReduce | ARBO | degree | k-clique | $O(m^{k/2})$ | $O(m+n)$ |
| Turán-shadow (*Jain & Seshadhri, 2017*) | random | | | | degeneracy | k-clique | $O(n\alpha^{k-1})$ | $O(n\alpha^{k-2}+m)$ |
| kClist (*Danisch, Balalau & Sozio, 2018*) | | enumeration | shared memory | ARBO | degeneracy | k-clique | $O(m\alpha^{k-2})$ | $O(m+P\alpha^2)$ |
| (*Eden, Ron & Seshadhri, 2018*) | rejection | | | | degree | k-clique | $\tilde{O}\left(\frac{n}{C_k^{1/k}} + \frac{m^{k/2}}{C_k}\right)$ | $O(m+n)$ |
| (*Li et al., 2020*) | | enumeration | shared memory | ARBO | color | k-clique | $O(km\frac{\Delta}{2}^{k-2})$ | $O(m+n)$ |
| Pivoter (*Jain & Seshadhri, 2020a*) | | counting | shared memory | Bron–Kerbosch | degeneracy | k-clique | $O(n\alpha3^{\alpha/3})$ | $O(m+n)$ |
| ARB-Count (*Shi, Dhulipala & Shun, 2021*) | color-based | enumeration | shared memory | ARBO | degeneracy | k-clique | exact: $O(m\alpha^{k-2})$ approximate: $O(pm\alpha^{k-2}+m)$ | $O(m+P\alpha)$ |
| YACC (*Jain & Tong, 2022*) | random | | | Turán-shadow | degeneracy | k-clique | $O(n\alpha^{k-1})$ | $O(n\alpha^{k-2}+m)$ |
| (*Yuan et al., 2022*) | | enumeration | shared memory | ARBO | degree/color | k-clique | $O(km\left(\frac{\Delta}{2}\right)^{k-2})$ | SDegree: $O(m+kN\Delta)$ BitCol: $O(m+N\frac{\Delta^2}{L})$ |
| (*Almasri et al., 2022*) | | counting | GPU | Bron–Kerbosch | degeneracy/degree | k-clique | not reported | $O(d^2{max})$ |
| DP-ColorPath (*Ye et al., 2023*) | color-based | | shared memory | | degeneracy | k-clique | k-color set: $O(\chi^k)$ k-color path: $O(\chi^{nk} + m)$ k-triangle path: $O(k\Delta)$ | k-color set: $O(m+n+\chi^k)$ k-color path: $O(kn + m)$ k-triangle path: $O(km)$ |
| EBBkC (*Wang, Yu & Long, 2024*) | | enumeration | shared memory | | color | k-clique | $O(md+k\cdot m\cdot\left(\frac{\tau}{2}\right)^{k-2})$ | $O(m+n)$ |
| SR-kCCE (*Chang, Gamage & Yu, 2024*) | random | | | | degeneracy | k-clique | not reported | not reported |
| BDAC (*Çalmaz & Bostanoğlu, 2024*) | without sampling | | | | degeneracy | k-clique | $O(\alpha^2)$ | $O(m+n+\alpha^2)$ |

**Notes.**

*These algorithms themselves are base algorithms.

that each vertex is processed more locally and efficiently, considering its neighbors. Building on this, *Dasari, Ranjan & Mohammad (2014)* developed the pbitMCE algorithm, which also leverages degeneracy ordering but diverges in its approach by using a partial bit adjacency matrix (pbam) to handle subgraphs. This data structure enhances vertex processing efficiency in a distributed computing environment like Hadoop, highlighting pbitMCE's suitability for large-scale graph data. The time complexity of the algorithm (*Eppstein, Löffler & Strash, 2010*) is affected by degeneracy, so it is suitable for sparse graphs with low degeneracy. The pbitMCE algorithm employs pivoting and uses parallelism to enhance the efficacy of Bron–Kerbosch-style algorithms; however, the resulting complexity remains exponential in $k$.

The paper MACE (*Makino & Uno, 2004*) proposes two strategies to list all maximal cliques but different than the Bron–Kerbosch, leveraging the matrix multiplication and maximum degree of the graph. This algorithm does not apply an ordering strategy and uses a depth-first backtracking procedure. The MACE algorithm has a similar complexity to ARBO (*Chiba & Nishizeki, 1985*). However, its efficiency decreases when the graphs get larger due to an additional factor, $n$. With a new strategy, the EBBkC (*Wang, Yu & Long, 2024*) presents better time complexity for k-clique listing than the ARBO, MACE, and kClist algorithms. It presents an edge-based branching strategy that explores larger cliques by adding connected vertex pairs. Besides, it also introduces three-edge sorting methods and early branch termination and incorporates parallelization techniques for improved performance over traditional vertex-based approaches.

To summarize, all the algorithms discussed so far are algorithms that count maximal cliques efficiently, using their own strategies and data structures for this purpose. These differences reflect trade-offs between memory usage, computational efficiency, and suitability for various computational environments. *Bron & Kerbosch (1973)* and *Akkoyunlu (1973)* emphasize direct enumeration, *Tomita, Tanaka & Takahashi (2006)* focus on memory-efficient structuring of the output, *Eppstein, Löffler & Strash (2010)* optimize through degeneracy ordering, and *Dasari, Ranjan & Mohammad (2014)* extend the strategy further with specialized data structures and distributed processing. While these algorithms use a similar approach to those proposed in Bron–Kerbosch, except the ones proposed in the paper MACE, they offer different strategies.

*Finocchi, Finocchi & Fusco (2015)* presents a MapReduce-based version of the ARBO algorithm and provides an approximate solution to relax the limitation of the exact approach. This algorithm complexity grows exponentially with $k$ but efficiently handles smaller cliques (lower $k$) in dense graphs.

Chiba and Nishizeki's ARBO (*Chiba & Nishizeki, 1985*) framework is improved by the kClist (*Danisch, Balalau & Sozio, 2018*) algorithm incorporating degeneracy ordering and parallelization techniques to enhance performance, particularly in handling large-scale graphs.

*Li et al. (2020)* presents a k-clique listing and counting approach based on the color orientation technique, which differs from the typical degree and degeneracy orientation methods. Similar to ordering-based k-clique algorithms such as kClist, this method deviates from the traditional degeneracy ordering approach. Instead, it employs color ordering to

list k-cliques within graphs effectively. *Li et al. (2020)* presents a decision tree to help select the most suitable k-clique listing algorithm based on different scenarios. This algorithm has similar time complexity with kClist but scales with maximum degree $\Delta$ instead of arboricity, making it suitable for sparse graphs. The SDegree and BitCol algorithms (*Yuan et al., 2022*) claim to have comparable time complexity and slightly better space efficiency than the algorithms proposed by *Li et al. (2020)*.

Exact k-clique counting algorithms depend on k-clique enumeration, which becomes infeasible for large graphs with high k values (*e.g.*, $k \geq 8$) due to combinatorial explosion. The Pivoter algorithm (*Jain & Seshadhri, 2020a*), inspired by the Bron–Kerbosch algorithm (*Bron & Kerbosch, 1973*), tackles this issue. The critical innovation of Pivoter is its ability to implicitly construct a succinct clique tree(SCT) using a pivoting technique during the search process. This SCT structure provides a unique and compact representation of all k-cliques, significantly reducing the space required compared to the total number of k-cliques. However, the authors of Pivoter note that there are certain graphs, such as com-lj (*Leskovec & Krevl, 2014*); even the parallel version of Pivoter struggled to count beyond $k = 10$. The Pivoter algorithm is also suitable for large and sparse graphs due to its time complexity, which depends on arboricity.

In response to the challenges of combinatorial explosion, there has been a shift towards approximation solutions using sampling methods. The Turán-shadow algorithm (*Jain & Seshadhri, 2017*) is the state-of-the-art sampling-based approximate k-clique algorithm for k values up to 10. This algorithm constructs a recursion tree, the Turán-shadow, to create dense subgraphs covering the entire graph, followed by an unbiased estimator to count the cliques. However, this process is time-intensive due to the construction of the shadow. Building on this approach, the YACC algorithm (*Jain & Tong, 2022*) reduces the recursion tree size to handle larger k values (up to 40) by relaxing the stopping condition during tree creation, improving efficiency but at the cost of accuracy. This reduction requires a significant increase in the number of samples to maintain reliable estimates. Both algorithms represent substantial advancements in approximate k-clique counting, balancing efficiency and accuracy through innovative techniques. The Turán-shadow algorithm is highly efficient for counting cliques in large, sparse graphs with low arboricity. Since the time and space complexity depends on the arboricity of the graph, it is suitable for sparse graphs. The YACC algorithm provides the same time and space complexity as Turán-shadow.

The algorithm ERS developed by *Eden, Ron & Seshadhri (2018)* is a randomized method for approximating the number of k-cliques in a given graph. ERS uses a query model, unlike the Turán-shadow algorithm (*Jain & Seshadhri, 2017*), which relies on constructing the Turán-shadow. ERS is more space-efficient compared to the memory-intensive Turán-shadow. However, while ERS theoretically achieves a $1 + \varepsilon$ approximation, its practical accuracy tends to be lower than that of Turán-shadow, as demonstrated in experiments reported by *Li et al. (2020)*. Additionally, *Li et al. (2020)* illustrates that for approximation algorithms, the worst-case time complexity of Turán-shadow is typically higher than that of ERS. Nonetheless, Turán-shadow's time overhead remains substantially lower than most exact algorithms.

The ARB-Count (*Shi, Dhulipala & Shun, 2021*) demonstrates it significantly outperforms the state-of-the-art parallel kClist algorithm and the parallel version of Pivoter. While Pivoter can handle all cliques in some large graphs, it could be more efficient for fixed k values and faces substantial slowdowns, particularly for smaller $k$. Moreover, Pivoter requires considerable memory and help with large graphs, often running out of space and failing to compute k-clique counts for higher $k$ values. In contrast, ARB-Count shows superior performance and efficiency, making it more suitable for practical use.

*Almasri et al. (2022)* compares GPU implementations with two CPU baselines: ARB-Count (*Shi, Dhulipala & Shun, 2021*), the top parallel graph orientation method, and Pivoter (*Jain & Seshadhri, 2020a*), the leading parallel pivoting method. Two key observations are presented. First, for small values of $k$, the graph orientation approach outperforms pivoting, which excels for larger $k$ values; this pattern holds for both CPU and GPU. Typically, the pivoting approach becomes superior around $k = 7$. Second, the best GPU implementation consistently outperforms the best parallel CPU implementation across all $k$ values. The ARB-Count and kClist (*Danisch, Balalau & Sozio, 2018*) algorithms are based on the ARBO (*Chiba & Nishizeki, 1985*) algorithm, and ARB-Count has better space complexity than kClist. The exact version is suitable for large and sparse graphs with low arboricity. The approximate version is suitable for large, sparse datasets where exact counting is infeasible.

*Ye et al. (2024)* proposes a framework for estimating the number of k-cliques by integrating Pivoter with three novel dynamic programming and color-based sampling techniques. The k-color set sampling algorithm's time and space complexities are affected by the number of colors used and $k$ values. It can be suitable for small $k$ values and graphs with low chromatic numbers ($\chi$). The k-color path sampling algorithm is suitable for small and moderate-size graphs and smaller $k$ values, as the complexity increases exponentially regarding $n$ and $k$ values. The complexity of the k-triangle algorithm depends on the number of triangles in the input graph and the desired clique size. This algorithm can suffer from scalability problems, especially for large, dense datasets with many triangles. This paper states that these sampling techniques outperform kClist and Pivoter across various datasets and $k$ values. Additionally, they note that the space overheads of their algorithms and Pivoter are comparable, while kClist consumes slightly more space than their methods.

The BDAC algorithm (*Çalmaz & Bostanoğlu, 2024*) establishes a boundary for the counts of k-cliques using existing extremal graph theorems. Notably, its complexity remains unaffected by the values of $k$. The authors demonstrate that BDAC is particularly well-suited for large and dense graphs, making it especially valuable for handling extensive datasets and larger $k$ values. Unlike most approximate algorithms, which rely on fixed-size samples and may not ensure sufficient sampling for large datasets, BDAC offers minimum and maximum k-clique counts. This provides guarantees grounded in theoretical foundations. However, this algorithm is not suitable for sparse datasets, failing to provide minimum clique counts and widening the gap between bounds.

The SR-kCCE (*Chang, Gamage & Yu, 2024*) algorithm provides an efficient approximate k-clique counting algorithm with guaranteeing accuracy. It does not specify explicit time

and space complexity but claims to generally outperform DP-ColorPath in execution time while being compatible with both Pivoter and DP-ColorPath regarding memory usage.

The algorithms whose time complexity depends on degeneracy or arboricity often perform well on large real-world graphs because the degeneracy and arboricity of a graph are much smaller than the maximum degree of the graph.

## CONCLUSION AND FUTURE DIRECTIONS

In this paper, we have explored the landscape of algorithms dedicated to counting k-cliques within a given graph *G*. Counting cliques is challenging because as the clique size *k* grows, the number of possible combinations increases exponentially. This survey analyses a wide range of methods that offer the solution for the clique counting problem, including detailed enumeration techniques, approximation strategies, and parallelization methods. The evolution of this algorithm from the past to the present, the strengths and weaknesses of the algorithms, and their limitations, if any, are detailed, then a comparative table is presented, highlighting their differences to guide future strategies.

Within this research's scope is a meticulously created taxonomy that systematically categorizes the approaches used in clique analysis. This taxonomy not only differentiates methodologies based on their precision—contrasting exact enumeration with approximate counting—but also includes parallelization strategies, delineating how computational tasks are distributed locally and globally.

A study proposed by *Li et al. (2020)* has explored and compared k-clique algorithms - particularly by introducing innovative heuristics to improve k-clique listing through greedy graph coloring - the primary focus has been developing specific techniques for pruning search paths. Our study provides a more comprehensive and well-organized overview of k-clique counting algorithms and covers a wider taxonomy covering exact, approximate, and parallelized techniques. The goal of classifying these methods is to make it easier for scholars to comprehend the state of k-clique counting and to recognize the advantages and disadvantages of the different strategies. It also stands out by incorporating more recent exact and approximate algorithms.

This paper addresses a significant gap in the literature by presenting a structured framework and in-depth analysis. It offers valuable insights and aims to foster a deeper comprehension of the methodologies used in k-clique counting research. The work establishes a solid foundation for future studies to develop more efficient and scalable algorithms for counting cliques in increasingly large and complex network structures.

Based on our thorough examination, we have identified several promising areas for future research. One potential improvement is incorporating advanced probabilistic models and machine learning algorithms to enhance the approximation methods. In the literature, there are articles that count graphlets using deep learning strategies (*Liu et al., 2018*; *Liu et al., 2021*). Still, no study has been found that employs machine learning, deep learning, or graph neural networks specifically for clique counting based on extracted features.

*Almasri et al. (2022)* utilizes the power of GPUs to accelerate parallel k-clique counting algorithms, including ARB-Count and Pivoter. These advancements demonstrate the

potential of GPUs to enhance the efficiency of k-clique counting significantly, providing faster solutions for large-scale graphs. There are also existing maximal clique counting algorithms optimized for GPU architectures that are not considered in this work (*Almasri et al., 2023*; *Wei, Chen & Tsai, 2021*). While GPUs offer significant potential to improve efficiency and provide faster solutions for large graphs, they also impose some limitations. A comprehensive benchmarking framework can be developed to evaluate the strengths and weaknesses of both GPU-based and CPU-based algorithms under different conditions. In addition, a hybrid approach that integrates CPU pre-processing with GPU acceleration can be developed to optimize resource usage and improve the efficiency of k-clique counting algorithms.

Additionally, only one algorithm currently provides local k-clique counts, which gives per-edge and per-vertex k-clique counts (*Jain & Seshadri, 2020a*). In addition, the BDAC (*Çalmaz & Bostanoğlu, 2024*) provides boundaries per vertices on large and dense graphs. Expanding on this approach could open up new opportunities for leveraging cliques in different contexts, such as clustering classification tasks, analyzing social and biological networks, and community detection/hiding. Using high-order cliques in these applications can provide different insights.

### Funding
The authors received no funding for this work.

### Competing Interests
The authors declare there are no competing interests.

### Author Contributions
- Büşra Çalmaz conceived and designed the experiments, analyzed the data, prepared figures and/or tables, authored or reviewed drafts of the article, and approved the final draft.
- Belgin Ergenç Bostanoğlu conceived and designed the experiments, authored or reviewed drafts of the article, and approved the final draft.

### Data Availability
This is a literature review.

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
