# Peer review of "k-Clique counting on large scale-graphs: a survey"

_PeerJ Computer Science, doi:10.7717/peerj-cs.2501_

## Round 0.1 · original submission · Major Revisions

I'd encourage authors to address all the major comments, especially from Reviewer 1 and 3, and revise the manuscript accordingly, with the exception of the Reviewer 3's comment on novelty and new algorithm (which is not applicable here given that it is a review paper).

Reviewer 1 ·

Basic reporting

- Clique counting is a relevant problem in CS and this paper fits the scope of the journal.

- The paper presents a comprehensive summary of existing literature in the area of clique counting. "Ordering Heuristics for k-clique Listing", published in VLDB 2020 also summarizes different clique counting algorithms. However, this paper includes more recent papers, adding information on algorithms that had not been published before 2020. The newer works are generally considered to be the current state of the art.

- The goals of the paper are listed clearly under the 'Research Questions' subsection.

Experimental design

- For the literature survey, papers are selected from a variety of credible sources.

- This paper includes all the relevant work, has the proper sources, and is organized appropriately.

- In addition to exact clique counting, related problems like approximating clique counts are also covered in detail. Various parallelization strategies present in these algorithms are described in the paper. Various graph ordering techniques are also discussed.

Validity of the findings

- Through the literature survey, the paper answers the research questions it lists in the beginning. It also provides different ideas for future work in this field. The short summaries for each algorithm is helpful for new readers looking to get familiar with clique counting.

- The paper claims that "comprehensive work has yet to review k-clique counting algorithms beyond triangles". This claim is not true since the aforementioned VLDB 2020 paper also reviews various clique counting algorithms (including ones reviewed in this paper). This paper differentiates itself by including newer exact and approximation algorithms.

- The paper does not spend enough time discussing the current state of the art algorithms. Including the pseudocode for ArbCount and Pivoter, which are the leading exact counting algorithms, would be more valuable to the reader. Additionally the paper presents a brute force algorithm twice (Algorithms 3 and 4) which are attributed to Jain and Seshadhri (2020), yet this algorithm is not present in the original paper. Furthermore the original work is not based on Arbo, but rather on Bron-Kerbosch.

Additional comments

- There are places throughout the paper where text is redundant. For e.g. on page 6, the summary of the two algorithms immediately following their descriptions is not needed.

- In some cases language/grammar can be improved.
For e.g. "The algorithm is the modified version of Jain (2020) enumerates all k-cliques in a given graph G." should be "The algorithm is the modified version of Jain (2020) which enumerates all k-cliques in a given graph G."

"A DAG is a graph in which it is impossible to follow a sequence of edges and return to the same vertex, thus forming a loop." - from this statement it may not be clear to an uninformed reader if DAGs are supposed to contain loops or not. Something like "A DAG is a graph in which loops do not exist, i.e. it is impossible to follow a sequence of edges and return to the same vertex." removes this ambiguity.

- In some cases, the reference does not match the paper which is referred to. Additionally. the reference format is not uniform throughout the paper.
"The PIVOTER has O(na3a/3) time complexity, n represents the number of vertices and a the degeneracy of the graph Ye et al. (2023)."

- Links to references in the main text would be helpful navigating the document.

Cite this review as
Anonymous Reviewer (2024) Peer Review #1 of "k-Clique counting on large scale-graphs: a survey (v0.1)". PeerJ Computer Science

Reviewer 2 ·

Basic reporting

The manuscript presents a comprehensive survey of k-clique counting, a topic that aligns well with the scope of the PeerJ Computer Science. It categorizes and examines various algorithms for k-clique counting, many of which have been recently surveyed in the literature. The Introduction section effectively sets the stage for the study, providing sufficient background on the topic. The manuscript offers a systematic analysis and comparison of both exact and approximation techniques, highlighting their respective advantages, limitations, and areas of suitability for different applications.

Experimental design

The survey methodology is explained in a detailed and comprehensive manner, and most sources are cited appropriately. However, it is important to note that Wikipedia is not considered a reliable source for academic research. The authors should replace the citation of Wikipedia with a reference to a relevant academic paper in the sentence: 'The presented Algorithm 1 (Wikipedia, 2024) finds all maximal cliques.' The survey provides a clear structure by starting with a definition of the methodology, followed by a step-by-step explanation of each type of algorithm, which enhances the clarity of the presentation.

Validity of the findings

As this manuscript is a survey paper, the Introduction and Conclusion sections appropriately summarize the overall flow and discuss the various categories of k-clique counting algorithms. The structure of each section is well-organized and clearly written, contributing to the coherence of the manuscript.

Additional comments

* Algorithms have been referenced beginning with capital letters or lowercase like "Algorithm 1" or "algorithm 2". Please check the paper template and correct them accordingly.

* "Graph arboricity" is not a commonly known term. It could be better to insert its definition in "Preliminaries" section.

Cite this review as
Anonymous Reviewer (2024) Peer Review #2 of "k-Clique counting on large scale-graphs: a survey (v0.1)". PeerJ Computer Science

Reviewer 3 ·

Basic reporting

- The paper contains weak English, leading to significant ambiguity. I have provided multiple comments in the attached PDF regarding this issue.

- Sections are not numbered leading to confusion in the paper structure.

Experimental design

Paper Overview
This paper surveys both exact and approximate k-clique enumeration and counting algorithms in the current literature, while also addressing parallel strategies for solving clique-related problems. It includes a summary table and explains a large body of work on maximal cliques and k-clique algorithms.
#####################

Paper Strengths
- It offers a summary of algorithms and techniques for enumerating and counting cliques, including maximal cliques and k-cliques.

- The discussion comprehensively covers enumeration and counting methods, both exact and approximate, as well as parallel strategies for existing clique algorithms.
##########################

Paper Weaknesses
- The analysis of k-clique counting algorithms omits discussions on memory complexity/requirements, evaluations on real datasets, large dataset analysis, and parallelization bottlenecks.

- The paper lacks a systematic approach for choosing an appropriate algorithm based on specific data or architecture.

- Although the paper claims to focus on k-cliques, it often conflates k-clique algorithms with those for maximal and maximum cliques. This creates confusion and detracts from the paper's clarity and focus.

- Table 2, meant to summarize the paper, is unclear as it mixes k-clique and maximal clique algorithms. Furthermore, the listed complexities are not specific to k-cliques.

- In the introduction, the authors claim that counting cliques larger than 3 is critical, but the evidence supporting this is insufficient. While they argue that larger cliques offer different insights, this claim is not adequately explained.

Validity of the findings

- The paper lacks significant novelty as its primary contribution is limited to a literature review of existing algorithms for k-clique and maximal clique counting. While it provides a comprehensive survey of existing methods, it does not introduce new algorithms, propose improvements, or offer fresh insights into the problem of clique enumeration. Furthermore, the paper does not present a novel framework or systematic approach for selecting appropriate algorithms based on specific datasets or computational architectures, nor does it delve into optimizing parallelization or memory usage. Therefore, its contribution remains largely in summarizing existing knowledge rather than advancing it.

- The complexity of the enumeration algorithms does not account for the complexity of storing the found cliques (e.g., using queues or other data structures), focusing instead on just reaching the cliques to consider them counted. As a result, the enumeration and counting discussed are merely algorithmic techniques for counting cliques, without considering the additional overhead of data management.

Additional comments

- Algorithm 1 should cite the official reference for the BK algorithm, rather than Wikipedia.

- The section on exact methods/enumeration mainly references papers on maximal clique enumeration.

- Table 2 also focuses on maximal clique algorithms, which are unrelated to k-clique counting. I suggest splitting the table into two separate tables for maximal and k-cliques.

- Since the GPU paper has both enumeration and counting capabilities (accelerating ARB-Count and PIVOTER on GPUs), it would be worth mentioning in this context.

- Additionally, given the discussion of maximal clique algorithms for CPUs, the implementation of maximal cliques on GPUs should also be highlighted.

Annotated reviews are not available for download in order to protect the identity of reviewers who chose to remain anonymous.
Cite this review as
Anonymous Reviewer (2024) Peer Review #3 of "k-Clique counting on large scale-graphs: a survey (v0.1)". PeerJ Computer Science

---

## Round 0.2 · accepted · Accept

Two reviewers are happy with the revision and recommended the acceptance of the article. As suggested by a reviewer, I'd also encourage the authors to take a careful editorial pass to improve the writing as the final step.

Reviewer 2 ·

Basic reporting

There are more recent algorithms that have been published recently and discussed in detail. Both Approximate and Exact Methods have been analyzed in more detail, and the authors' recently published work has been compared to the state-of-the art approaches. The Introduction section has been expanded to include a discussion on larger cliques.

Experimental design

The Survey methodology has been categorized into Exact, Approximate and Parallel Strategies, where the methodolgical approach is appropriate. Parallelization strategies, particularly GPU- oriented algorithms, have not been thoroughly explained in recent studies. However, this section has been expanded to include a discussion of Almasri et al.'s work. The overall discussion in the previous version was insufficient. In the revised version, more comparisons are provided, and the time/space complexities of the reviewed approaches are cleary presented

Validity of the findings

The Introduction secton offers surveying various k-clique counting approaches, discussing the prevalence of higher numbers of cliques across different domains. This topic has been elaborated on within the text. The Conclusion has also been expanded to include GPU-oriented approaches. Additionally, it would be beneficial to suggest some future research directions, particularly for parallelization approaches, where advancements in processing unit technologies are emerging.

Cite this review as
Anonymous Reviewer (2024) Peer Review #2 of "k-Clique counting on large scale-graphs: a survey (v0.2)". PeerJ Computer Science

Reviewer 3 ·

Basic reporting

### Basic Reporting
The revised paper significantly improves its English and adds numerous clarifications to resolve ambiguities compared to the first version. I have provided additional comments in the attached PDF on how to enhance the writing further.
- The lack of section numbering leads to some confusion regarding the structure of the paper.

Experimental design

#### Paper Overview
The paper provides a comprehensive survey of both exact and approximate k-clique enumeration and counting algorithms in current literature, while also discussing parallel strategies for solving clique-related problems. It includes a summary table and presents a substantial body of work on maximal cliques and k-clique algorithms.

#### Paper Strengths
- The paper offers a detailed summary of algorithms and techniques for enumerating and counting cliques, including maximal cliques and k-cliques.
- The discussion thoroughly covers both exact and approximate enumeration and counting methods, as well as parallel strategies for existing clique algorithms.

#### Improvements in the Revised Paper
- The paper now clearly states early on that it does not analyze 3-cliques, bi-cliques, quasi-cliques, or maximal cliques unrelated to clique counting, which helps prevent confusion for readers.
- Time and memory complexity are now explicitly mentioned for many of the algorithms in the text.
- The discussion now includes recent works, such as Yuan et al. (2022) and Wang et al. (2024), which further enrich the paper.
- Additional discussion on parallelism bottlenecks for clique counting on shared memory systems and GPUs has been added, which is a critical inclusion.
- Maximal cliques on GPUs are mentioned.
- Table 2 has been revised and is much clearer and more informative in this version.

#### Revised Paper Weaknesses
- All of my previous comments have been addressed in this version of the paper.
- The authors should conduct a final review to improve the English throughout the text.
* * *
Validity of the findings

- The revised paper covers a broad spectrum of algorithms and approaches to clique counting and highlights the novelty of clique counting and enumeration.
- It also includes an up-to-date list of references, creating a comprehensive and current literature review.

Additional comments

I would like to sincerely thank the authors for addressing all my comments and significantly improving the quality of the paper. The paper now clearly states its objectives, includes time and space complexity details, incorporates the latest relevant studies, enhances the clarity of Table 2, discusses bottlenecks across different computing architectures, and mentions the role of maximal cliques on GPUs.


I believe the paper is ready for publication, pending a final review of the English for clarity and polish.

Annotated reviews are not available for download in order to protect the identity of reviewers who chose to remain anonymous.
Cite this review as
Anonymous Reviewer (2024) Peer Review #3 of "k-Clique counting on large scale-graphs: a survey (v0.2)". PeerJ Computer Science